# Functional Characterization of Three Diterpene Synthases Responsible for Tetracyclic Diterpene Biosynthesis in *Scoparia dulcis*

**DOI:** 10.3390/plants12010069

**Published:** 2022-12-23

**Authors:** Jung-Bum Lee, Tomoya Ohmura, Yoshimi Yamamura

**Affiliations:** Graduate School of Medicine and Pharmaceutical Sciences, University of Toyama, 2630 Sugitani, Toyama 930-0194, Japan

**Keywords:** diterpene synthase, *Scoparia dulcis*, cyclase

## Abstract

*Scoparia dulcis* produces unique biologically active diterpenoids such as scopadulcic acid B (SDB). They are biosynthesized from geranylgeranyl diphosphate (GGPP) via *syn*-copalyl diphosphate (*syn*-CPP) and scopadulanol as an important key intermediate. In this paper, we functionally characterized three diterpene synthases, SdCPS2, SdKSL1 and SdKSL2, from *S. dulcis.* The SdCPS2 catalyzed a cyclization reaction from GGPP to *syn-*CPP, and SdKSL1 did from *syn-*CPP to scopadulan-13α-ol. On the other hand, SdKSL2 was found to incorporate a non-sense mutation at 682. Therefore, we mutated the nucleotide residue from A to G in SdKSL2 to produce SdKSL2mut, and it was able to recover the catalytic function from *syn*-CPP to *syn*-aphidicol-16-ene, the precursor to scopadulin. From our results, SdCPS2 and SdKSL1 might be important key players for SDB biosynthesis in *S. dulcis*.

## 1. Introduction

Plants produce a vast array of secondary metabolites, and these natural products are of increasing importance as a resource for drugs and drug leads. The chemical diversity of natural products is enormous, and is derived from its biosynthetic machinery. Among important natural products, terpenes are the largest group, and they range from simple flavor and fragrance compounds to complex triterpenoids. Large populations of terpenes are cyclic compounds possessing several chiral centers, and the cyclization catalyzed by terpene synthases is the first step in constructing their diverse structures. Scopadulcic acid B (SDB, **1**) is a unique tetracyclic diterpenoid produced by a perennial medicinal herb, *Scoparia dulcis* L. (Plantaginaceae, Lamiales), which is distributed among tropical and subtropical regions [1]. SDB, **1** has attracted substantial attention both for its multiple biological effects and its unique tetracyclic skeleton. Early studies suggested that the SDB biosynthetic process was initiated by two sequential steps: the formation of *syn-*copalyl diphosphate (*syn*-CPP, **3**) from geranylgeranyl diphosphate (GGPP, **2**), and the subsequent cyclization of the tetracyclic skeleton from **3** (Figure 1). Such dual cyclization schemes are characteristic machinery for the biosynthesis of labdane-related diterpenoids (LRDs) [2,3].

Recently, we have discovered a gene candidates responsible for the biosynthesis of **1** by the RNA-seq analysis of *S. dulcis* [4]. In this study, we describe the characteristics of the key enzymes for unique diterpene biosynthesis in *S. dulcis.*

## 2. Results

### 2.1. Cloning of Diterpene Synthases

Gene candidates for unique diterpene synthases (DTSs) were cloned from a cDNA library prepared with *S. dulcis*. Both gene candidates were cloned from the cDNA library prepared with *S. dulcis*. *SdCPS2* (2412 bp) encoded 804 amino-acid residues containing the *N-*terminal transit-peptide sequence. It showed typical features of monofunctional-angiosperm class II diterpene synthases (DTSs) featuring a three-domain structure (γ-β-α). It also included the characteristic DxDD motif, which functions as the general acid to protonate the terminal carbon-carbon double bond of the substrate **2**. In addition, SdCPS2 contained SAYDTAW and QxxDGSW motifs, which are well-conserved in class II DTSs (Appendix A) [5]. The function of the former motif is still unknown, whereas the later motif is proposed to be involved in the stabilization of the intermediate cation during the cyclization. On the other hand, *SdKSL1* (2412 bp) encoded 805 amino acid residues with *N*-terminal transit-peptide sequence. Moreover, SdKSL1 also featured a three-domain structure the same as SdCPS2. SdKSL1, containing the characteristic DDxxD and NSE/DTE motifs for class I DTSs (Appendix A). These conserved motifs are involved in metal-dependent ionization of the prenyl diphosphate substrates. In addition, we cloned another class I DTS gene, *SdKSL2* (2401 bp). The coding sequence (CDS) of *SdKSL2* covered nearly the full length of the gene when compared with those of class I DTSs. It was noteworthy that the CDS introduced a non-sense mutation at 682 position (Appendix A). The identity between SdKSL1 and SdKSL2 was very high (88%).

Homology searches against protein sequences indicated that SdCPS2 showed high identity with functionally annotated *syn*-CPS, VacTPS3 from *Vitex angus-castus* (AUT77122, 63%) [6]. In addition, it is noteworthy that identity against *syn*-CPS from *Oryza sativa* (OsCPS4, BAD42451, 39%) [7] is quite low. On the other hand, SdKSL1 showed extended similarities with already functionally annotated KSLs from *Isodon rubescens* such as IrKSL6 (isopimaradiene synthase, ASC55318, 46%) [8] and IrTPS2 (nezukol synthase, ARO38140, 46%) [9]. So far, VacTPS6 from *V. angus-castus* (*syn*-isopimara-7,15-diene synthase, AUT77125) [6] have been shown to accept *syn*-CPP as a substrate; however, the identity between SdKSL1 and VacTPS6 was 44%. Finally, SdKSL2 showed the highest identity against putative *cis*-abienol synthase from *Handroanthus impetiginosus* (PIN17551, 50%).

As shown in Figure 1, phylogenetic relationships were found for diterpene synthases. In particular, SdCPS2 was placed into the clade, consisting of class II DTSs responsible for secondary metabolism. It is noteworthy that SdCPS2 was found to be closely related to VacTPS3, which is reported to be a *syn*-copalyl diphosphate synthase. On the other hand, SdKSL1 and SdKSL2 were also placed into the clade consisting of class I DTSs related to the secondary metabolism. Therefore, these three diterpene synthases might be involved in unique diterpene biosynthesis in *S. dulcis*.

### 2.2. Characterization of Enzymic Function of DTSs

In order to characterize the enzymatic function of SdCPS2, cDNA was truncated to remove signal-sequence targeting for the plastid, and then ligated into the pSdGG vector to construct pSdGG/SdCPS2 plasmid. This plasmid harbors GGPP synthase from *S. dulcis* to provide the substrate GGPP for SdCPS2 in recombinant *E. coli*. So far, transit-peptide sequences have been shown to interfere functionally with the expression of diterpene synthases; therefore, we truncated the corresponding nucleotide sequences to construct the recombinant cells. Together with the construct, transformants with pSdGG/SdCPS1 (*ent*-CPS from *S. dulcis*), pSdGG/SmCPS1 (*normal-*CPS from *Salvia miltiorrhiza*) [10] or pSdGG/OsCPS4 (*syn*-CPS from *Oryza sativa*) [11] were prepared in order to compare the enzymatic reaction products. The recombinant cells were cultured in the Terrific-broth liquid media, supplemented with 1% glucose and induced with 0.5 mM IPTG. After 72 h incubation at 16 °C, the medium was extracted twice with *n*-hexane, and then the extract was concentrated and analyzed using GC-MS. In our assay system, we confirmed that diterpene alcohols as dephosphorylated products were secreted to culture media from host cells.

*E. coli* harboring pSdGG/SdCPS2 produced a single diterpene hydrocarbon as peak **3′** (Figure 2A), and its retention time and mass spectra were identical with that derived from OsCPS4 (Figure 2B). In addition, different retention times of the enzymatic products **5′** (Figure 2C) and **6′** (Figure 2D) were observed in the case of SdCPS1 (*ent-*CPS) and SmCPS1 (*normal*-CPS), respectively. Therefore, SdCPS2 was judged not to be *ent*- or *normal-*CPS. To provide further evidence for the identity of the SdCPS2 product, the diterpene alcohol derivative was isolated using silica gel column chromatography and applied to the NMR analyses (Appendix A). As a result, its structure was found to be identical to a *syn*-copalol, based on the comparison data from the ^1^H- and ^13^C-NMR chemical shifts (Appendix A) [12]. However, it is impossible to separate enantiomers such as *syn*-copalol and *syn-ent-*copalol on the GC-MS equipped with an achiral-phase capillary column, and we were not able to achieve fine separation using chiral GC (data not shown). For this reason, we chose to utilize a known enzyme (OsKSL4) for cyclization from *syn-*CPP to *syn-*pimara-7,11-diene (**7**) [13]. Recombinant *E. coli* transformed with pSdGG/SdCPS2 and pOsKSL4 was cultured and treated with IPTG to produce **7,** as shown in Figure 2E. The retention time and mass spectrum were identical to those produced by the recombinant *E. coli* transformed with pSdGG/OsCPS4 and pOsKSL4 (Figure 2F). This indicated that the enzymic-reaction products of SdCPS2 could be cyclized by OsKSL4, whose substrate is a *syn*-copalyl diphosphate. Therefore, our cloned SdCPS2 was found to be a *syn*-CPS.

Subsequently, pSdKSL1 and pSdKSL2mut were constructed, and transformed with pSdGG/SdCPS2 into *E. coli* cells to elucidate the function of both DTSs. As described above, *SdKSL2* contained a non-sense mutation in its CDS. By comparing the nucleotide sequences of *SdKSL1* and *SdKSL2*, we introduced a mutation (A682G) to recover its enzymatic function (*SdKSL2mut*). As shown in Figure 3A, peak **4′** appeared together with peak **3′**, which corresponded to a *syn*-copalol. The mass spectrum of peak **4′** revealed that the corresponding compound was suggested to be a novel diterpene alcohol, synthesized by SdKSL1. In order to identify this enzymatic-reaction, product **4** was isolated using silica gel column chromatography from large-scale culture. NMR analyses (Appendix A), including 2D experiments of **4,** revealed that it was identical to scopadula-13α-ol (demalonyl thyrsiflorin A) (Appendix A) [14]. Therefore, SdKSL1 could be regarded as a novel class I DTS, scopadula-13α-ol synthase. On the other hand, SdKSL2mut was able to produce different diterpene hydrocarbons, as shown in Figure 3B. The mass spectrum of peak **8** was identical to that of a previously reported aphidicol-16-ene [13,15]. In order to confirm its chemical structure, we purified it using silica gel column chromatography, and applied it to NMR analyses (Appendix A). The ^1^H and ^13^C NMR chemical shifts of **8** were identified as an aphidicol-16-ene (Appendix A).

### 2.3. Quantification of Scopadulcic Acid B (1) and Realtime qPCR Analysis of SdCPS2 and SdKSL1

As described above, *SdCPS2* and *SdKSL1* were suggested to be involved in the biosynthesis of unique tetracyclic diterpenes in *S. dulcis*. Therefore, we analyzed the amounts of scopadulcic acid B (**1**) and the expression levels of those transcripts in plant tissues. As shown in Figure 4A, scopadulcic acid B (**1**) accumulated in the young leaf. Similarly, the highest expression levels of *SdCPS2* and *SdKSL1* were observed in the young leaf (Figure 4B).

## 3. Discussion

In this report, we discovered three diterpene synthases responsible for the unique tetracyclic diterpene biosynthesis in *S. dulcis*. Reaction mechanisms of SdKSL1 and SdKSL2 are proposed, as shown in Figure 2. At first, the lysis of the allylic diphosphate ester in **3** induces type A cyclization, initiated by the formation of an allylic cation, followed by *si-*face attack of the olefin, which is accompanied by a hydride shift from C-9 to C-8 to afford the 8β-pimarenyl cation (**10**) [5]. The carbocation is suggested to cyclize with the vinyl group to give **11**, and rearrangement of **11** provides secondary cation **13,** via tertiary cation **12**. This undergoes trapping with the nucleophilic oxygen of water, to give tetracyclic diterpene alcohol **4**. On the other hand, SdKSL2 is suggested to produce an aphidicol-16-ene (**8**). *S. dulcis* has been reported to produce an aphidicolane-type diterpene, scopadulin [16]. Thus, the functionally active SdKSL2-related enzyme may be involved in the biosynthesis of scopadulin.

Figure 4 reveals that the preserved tissue, i.e., young leaf, of scopadulcic acid B (**1**) corresponded to the expression levels of *SdCPS2* and *SdKSL1*. The data might strongly support these two DTSs being involved in the unique diterpene biosynthesis.

In conclusion, it is shown here that SdCPS2 and SdKSL1 exhibit class II and class I diterpene synthase activities, respectively. The former catalyze the formation from **2** to **3**, and the later cyclize **3** to **4**; therefore, SdCPS2 and SdKSL1 were regarded as *syn-*CPS and scopadula-13α-ol synthase, respectively. So far, two *syn*-CPSs, OsCPS4 and VacTPS3, have been reported, whereas this paper is the first to report the unique tetracyclic diterpene cyclase, SdKSL1. Although **4** was not isolated from natural sources, a structurally related one, thyrsiflorin A, was isolated from *Calceolaria thyrsiflora*. During structural elucidation, demalonyl thyrsiflorin A was reported, and it was found that the compound was identical to **4**. Recently, **4** has been reported to have cytotoxic effects on melanoma cells [17] and inducible effects on apoptosis and necrosis in the human cancer cell-line [18]. In these papers, **4** was prepared by the reduction of naturally occurring thyrsiflorin A. Therefore, the cloned enzymes described in this study, under large-scale fermentation production could provide a way to produce **4** for further evaluation of its biological effects.

## 4. Material and Methods

### 4.1. Plant Material

*Scoparia dulcis* was germinated under sterile conditions and was grown on half-strength Murashige and Skoog agar medium at 25 °C, under continuous illumination. After 5–6 weeks of growth, the seedlings were harvested, frozen immediately in liquid nitrogen, and stored at −80 °C for RNA isolation.

### 4.2. Cloning of Diterpene Synthases

The total RNA was isolated from the seedlings using TRIzol reagent (Invitrogen, Carlsbad, CA, USA) and cDNA was generated by the reverse-transcription reaction, using the PrimeScript II First-strand cDNA synthesis kit (Takara Bio Inc., Kusatsu, Shiga, Japan). The *SdCPS2*, *SdKSL1*, and *SdKSL2* genes were cloned, using the primers described.

### 4.3. Construction of Expression Vectors

All expression vectors were constructed according to the method previously reported by Cyr et al. [19]. Briefly, *SdGGPPS* (Accession No. AB034250) was truncated, to remove the transit-peptide sequence (57 amino acids), and introduced into the pACYC-Duet (Novagen Merck, Darmstadt, Germany) multiple cloning site 2 (MCS2). The ORF of *SdCPS2*, after the transit-peptide sequence was truncated, was then cloned into the MCS1 of the vector, using the corresponding *Nco*I and *Not*I restriction site to construct pSdGG/SdCPS2. Similarly, *SdCPS1* (Accession No. AB169881), *SmCPS1* (Accession No. KC814639), and *OsCPS4* (Accession No. BAD42451) were cloned into MCS1, to give pSdGG/SdCPS1, pSdGG/SmCPS1, and pSdGG/OsCPS4 vectors, respectively. *SdKSL1* or *OsKSL4* (Accession No. Q0JEZ8) were cloned into the multiple cloning site in pET28b (Novagen Merck), using the corresponding *Nco*I and *Not*I restriction site, to give pSdKSL1 or pOsKSL4 plasmids. The *SdKSL2* contained the stop codon.

### 4.4. Heterologous Expression and Metabolite Analysis

The resulting constructs were transformed into the C41 strain of *E. coli* (Lucigen), and heterologously expressed. These recombinants were grown in liquid TB media (12 g casein, 24 g yeast extract, 8 mL 50% glycerol in 1 L H_2_O, and pH adjusted to 7.0) with the appropriate antibiotics at 37 °C to OD_600_~0.6, then transferred to 16 °C for an hour and induced with IPTG (0.5 mM final concentration). After 24 h growth at 16 °C, the dephosphorylated enzymatic products were extracted with *n-*hexane (equal volume). The organic layer was separated out and then dried under N_2_, with the residue re-suspended in fresh hexanes and analyzed using GC-MS. Note that dephosphorylation of the class II diterpene synthase products, **3**, **5**, and **6**, to generate **3′**, **5′**, and **6′** are mediated by the endogenous phosphatases in *E. coli* as described elsewhere [20]. GC-MS analysis was carried out using a DB-5MS fused-silica capillary column (Agilent) on a Shimadzu QP-2010 *Ultra* gas chromatograph mass spectrometer in electro-ionization mode (70 eV). Each sample was injected at 230 °C in the splitless mode. The samples were initially held at 50 °C for 1 min, following which the oven temperature was increased by 10 °C/min to 300 °C and held for 5 min. The flow rate of He carrier gas was set at 1.9 mL/min. The MS data were collected from 40 to 400 *m/z*.

### 4.5. Phylogenetic Analyses

Phylogenetic analyses were performed using RAxML software [21] with alignments prepared using the MAFFT program. Selected DTS proteins were aligned by employing a highly accurate method: L-INS-I. The substitution model was selected according to the maximum AIC, as determined by ModelTest-NG [22]. Maximum-likelihood (ML) trees were built, using the JTT+I+G4+F model and 1000 replicates of the bootstrap analyses, and the obtained phylogeny was displayed using FigTree software (http://tree.bio.ed.ac.uk/software/figtree) (accessed on 18 November 2018).

### 4.6. Quantifation of Scopadulcic Acid B

The quantification of scopadulcic acid B was performed, using the previously reported method [23]. Briefly, 1 mL of CHCl_3_:MeOH (3:1) solution was added to the freeze-dried young leaf (YL), mature leaf (LE), stem (ST), and root (RO) (approximately 20 mg) of *S*. *dulcis,* and the mixture was sonicated for 20 min. After centrifugation at 13,000 rpm for 10 min, 1 mL of distilled water was added to the supernatant, and the mixture was vigorously shaken at room temperature for 30 min. After centrifugation at 13,000 rpm for 10 min, 0.6 mL of the lower layer (CHCl_3_ layer) was dried in vacuo. The residue was dissolved with 1 mL of MeOH and quantified using the HPLC system (Hitachi, D7000 system) equipped with a Cosmosil C18ARII column (Nacalai Inc., Kyoto, Japan).

### 4.7. Real-Time qPCR Analysis

Real-time qPCR was performed using Brilliant III Ultra-Fast SYBR Green QPCR Master Mix with ROX (Agilent Technologies, Santa Clara, CA, USA) on an AriaMX real-time QPCR system (Agilent Technologies). The *S*. *dulcis* 18S rRNA gene was used for normalization. The primer sequences used in the qPCR study are listed in Appendix A. Calibration curves were produced for each of the primer pairs, and quantification was performed using the Agilent AriaMx software (Agilent Technologies, ver.1.8). Each sample was tested three times.

## Data Availability

The data contained within the present article and in its Appendix A are freely available upon request to the corresponding author.

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
