# Peer review of "Functional Characterization of Three Diterpene Synthases Responsible for Tetracyclic Diterpene Biosynthesis in Scoparia dulcis"

_plants, 2022, doi:10.3390/plants12010069_

Round 1
Reviewer 1 Report
The manuscript entitled “Functional Characterization of Three Diterpene Synthases Responsible for Tetracyclic Diterpene Biosynthesis in Scoparia 3 dulcis” by Lee and coworkers have very clearly described the role of 3 diterpene synthases, viz., SdCPS2, SdKSL1 and SdKSL2 in the production of scopadula-13a-ol (referred to as Compound 4). The study also further provides a way to produce this compound in large scale in a culture broth. The manuscript provides new and significant information accordingly should be published. While the paper has been written fairly well, some revision of English language would make it read very well. For example, here are two key sentences I have rewritten.
Line 42: Here, In this study, we showed describe the characteristics of key enzymes for unique diterpene biosynthesis in S. dulcis.
Line 189: Therefore, our the cloned enzymes, described in this study, under large scale fermentation production could provide a way to produce 4 for further evaluation of its biological effects.
Kindly have the paper revised to improve its readability.
Overall it is an excellent study.
Author Response
- We correct the english writing suggested by the reviewer 1. "Here, we showed ..." to "In this study, we describe ..." (L42)
- We revised the english writing suggested by the reviewer 1. " Therefore, the cloned enzymes, described in this study, under large scale fermentation production could provide a way to produce 4 for further evaluation of its biological effects." (L189)
Reviewer 2 Report
This manuscript describes investigation of diterpene synthases/cyclases from the medicinal herb Scoparia dulcis, leading to functional identification of the consecutively acting class II diterpene cyclase SdCPS2 and class I diterpene synthase SdKSL1 almost certainly involved in biosynthesis of the bioactive diterpenoid scopadulcic acid. SdCPS1 converts the linear general diterpene precursor GGPP to bicyclic syn-CPP, while SdKSL1 converts this to the more specific precursor scopadulanol. Several (syn-)CPP producing CPSs have been previously identified, while SdKSL1 appears to be novel in its production of the highly rearranged tetracyclic scopadulanol. Overall the work seems solid, but the description suffers from some deficiencies. There are numerous grammatical errors throughout, only some of which are described below. More troublingly, there is a lack of scholarship – i.e., appropriate referencing. For example, while references for the two previously identified syn-CPP synthases are provided (and that for OsCPS4 does not appear to be actually cited in the text), there appear to be some missing for at least some of the class I diterpene synthases mentioned in the text (e.g., IrTPSKSL6, IrTPS2 and HiCAS, although it is not clear that the last has actually been functionally characterized). Indeed, references and GenBank accession information should be provided for all the diterpene synthases used for the shown phylogenetic analysis (Figure 1), at least in the supplemental information. In addition, reference(s) should be provided for the catalytic motifs noted here – at least a review article such as one of those by Prof. Christianson (e.g., Chemical Reviews, 2017), if not the primary literature (e.g., for the function of the DxDD motif as a general acid – Prisic et al., 2007, ChemBioChem 8, 869). Perhaps related to this, the description of the class I TPS catalyzed reaction needs to be corrected (line 165). These are not dephosphatases, but rather catalyze lysis of the allylic diphosphate ester bond in their substrates to form an allylic carbocation.
A more moderate issue arises from the “rescue” of SdKSL2, by correction of the native premature stop codon, which reveals this to be an aphidicolene synthase. Does this have any physiological relevance – i.e., are any derived diterpenoid natural products found in S. dulcis? Also, what does “identity” refer to in this study – i.e., is this amino acid or nucleotide sequence based? All of these points must be clarified.
Figure 1: “SmKS” almost certainly should be “SmKSL”.
Figure 2: Legend needs clarify that CPSs produce CPP, which are then dephosphorylated by endogenous phosphatases, leading to the primary alcohol derivatives observed by GC-MS.
Scheme 2: The hydride undergoing a 1,2-shift in the transformation from 3 to 9 needs to be shown (actually might be best to show the initially formed pimar-13-en-8-yl carbocation).
Minor corrections:
Ln 80: “found diterpene” should be “found for diterpene”
Ln 82: “to closely related VacTPS3” should be “to be closely related to VacTPS3”
Ln 91: “plasmid is harboring GGPP” should be “plasmid harbors GGPP”
Ln 138: “4’ was appeared” should be “4’ appeared”
Ln 144: “class II DTS” should be “class I DTS”
Ln 183: “resources” should be “sources”
Author Response
- We added references [19-22] as suggested by the reviewer 2. In addition, we inserted the term "putative" into the L73, because function of cis-abienol synthase (PIN17551) is still unknown.
- We added reference [23] which describes the motif of diterpene synthases, as suggested by the reviewer. (L55). In addition, we corrected the sentence at L165 as suggested by the reviewer.
- We added the sentence "S. dulcis has been reported to produce an aphidicolane-type diterpene, scopadulin [24]. Thus, the functionally-active SdKSL2 related enzyme may be involved in the biosynthesis of scopadulin." by answering the request by the reviewer.
- We performed homology searches of DTSs against protein sequences. Thus, we added to the phrase "against protein sequences" was inserted in L65.
- We prepared Table S2. which is summarized abbreviations and accession numbers of DTSs used in Figure 1. We added references as suggested by the reviewer.
- Figure 1. corrected from SmKS to SmKSL1 as suggested by the reviewer.
- We corrected caption of Figure 2 as suggested by the reviewer.
- We revised scheme 2 as suggested by the reviewer.
Minor correction
- We inserted "for" after found as suggested by the reviewer. (L82)
- We revised the sentence from "to closely related" to "to be closely related to" as suggested by the reviewer. (L84)
- We revised the word from "is harboring" to "harbors" as suggested by the reviewer (L93)
- We removed "was" in L143 as suggested by the reviewer.
- We corrected "class II DTS" to "class I DTS". (L149)
- We changed "resources" to "sources" as suggested by the reviewer (L190)
Round 2
Reviewer 1 Report
Authors have made appropriate changes and the manuscript is acceptible.
Author Response
We thank the reviewer's kind suggestions. We believe that our manuscript was well improved by the reviewer's comments.
Reviewer 2 Report
This manuscript has been suitably revised to address some but certainly not all of my previously expressed concerns. Among those unaddressed is the nature of the reaction catalyzed by the KSLs (class I not type A TPSs). At a minimum, on line 173 “dephosphorylation of” should be replaced with “lysis of the allylic diphosphate ester in”. Similarly, in description of the characterization of CPS activity in the legend of Figure 2, it is implied that these produced dephosphorylated copalol rather than their actual copalyl diphosphate products. At a minimum, all three instances of “dephosphorylated x (x-copalol, x’) by XxCPS” should be replaced by “x by XxCPS, observed as the dephosphorylated derivative x-copalol (x’) produced by endogenous phosphatases”.
There are two other changes that would help improve appreciation of the significance of SdKSL2 as well as clarified the catalyzed reaction. First, note the derivation of scopadulin from syn-aphidicolene in the abstract. In particular, on line 16 after “syn-aphidicol-16-ene” add “, the precursor to scopadulin”. Second, in Scheme 2 please annotate the arrow over “SdKSL1” with an “a”, and that next to “SdKSL2” with a “b” (to further clarify the alternative mechanisms shown in the preceding intermediate).
Author Response
Thank you for your kind suggestion. We apologize our mistake to answer the reviewer's kind suggestion. We corrected carefully our manuscript as suggested by the reviewer. We hope that our revised manuscript will be satisfied the review's recommendation.
- We revised the term "dephosphorylation of" to "lysis of the allylic diphosphate ester in" as indicated by the reviewer. (L222) In addition, we revised the caption of Figure 2 as indicated by the reviewer.
- We revised in Abstract as suggested by the reviewer, e.g., "syn-aphidicol-16-ene, the precursor to scopadulin." (L16)
- We add the "a" and "b" over arrow in Scheme 2, as suggested by the reviewer.